


**Tropospheric ozone and its precursors at Summit, Greenland: comparison between**
**observations and model simulations**
Yaoxian Huang[1,a], Shiliang Wu[1,2,3], Louisa J. Kramer[1,2,b], Detlev Helmig[4], and Richard E.
Honrath[1,2,†]
[1]Department of Geological and Mining Engineering and Sciences, Michigan Technological
University, Houghton, Michigan, USA
[2]Atmospheric Sciences Program, Michigan Technological University, Houghton, Michigan,
USA
[3]Department of Civil and Environmental Engineering, Michigan Technological University,
Houghton, Michigan, USA
[4]Institute of Arctic and Alpine Research, University of Colorado, Boulder, Colorado, USA
[a]now at: School of Forestry and Environmental Studies, Yale University, New Haven,
Connecticut, USA
[b]now at: University of Birmingham, Birmingham, UK
[†]deceased
*Correspondence to*: S. Wu (slwu@mtu.edu) and Y. Huang (yaoxian.huang@yale.edu)
**Abstract.** Recent studies have shown some significant challenges for atmospheric models to
simulate tropospheric ozone ($O_3$) and some of its precursors in the Arctic. In this study, ground
based data are combined with a global 3-D chemical transport model (GEOS-Chem) to examine
the abundance and seasonal variations of $O_3$ and its precursors at Summit, Greenland (72.34° N,
38.29° W, 3212 m.a.s.l). Model simulations for atmospheric nitrogen oxides ($NO_x$), peroxyacetyl
nitrate (PAN), ethane ($C_2H_6$), propane ($C_3H_8$), carbon monoxide (CO), and $O_3$ for the period of
07/2008-06/2010 are compared with observations. The model performs well in simulating certain
species (such as CO and $C_3H_8$), but some significant discrepancies are identified for other
species and further investigated. The model generally underestimates $NO_x$ and PAN (by around
50% and 30%, respectively) for March-June. Likely contributing factors to the low bias include
missing $NO_x$ and PAN emissions from snowpack chemistry in the model. At the same time, the
model overestimates $NO_x$ mixing ratios by more than a factor of 2 in wintertime, with episodic
$NO_x$ mixing ratios up to 15 times higher than the typical $NO_x$ levels at Summit. Further





investigation shows that these simulated episodic $NO_x$ spikes are always associated with
transport events from Europe, but the exact cause remains unclear. The model systematically
overestimates $C_2H_6$ mixing ratios by approximately 20% relative to observations. This
discrepancy can be resolved by decreasing anthropogenic $C_2H_6$ emissions over Asia and the US
by ∼ 20%, from 5.4 to 4.4 Tg/yr. GEOS-Chem is able to reproduce the seasonal variability of $O_3$
and its spring maximum. However, compared with observations, it underestimates surface $O_3$ by
approximately 13% (6.5 ppbv) from April to July. This low bias appears to be driven by several
factors including missing snowpack emissions for $NO_x$ and nitrous acid, the coarse model
resolution, model overestimated $O_3$ dry deposition velocity during springtime, as well as the
uncertainties in the stratosphere-to-troposphere exchange scheme for $O_3$.

## 1. Introduction

Ozone ($O_3$) and its precursors (e.g., $NO_x$ = NO + $NO_2$ and volatile organic compounds) are
important atmospheric species affecting both air quality and climate (e.g., Jacob et al., 1992;
Fiore et al., 2002; Unger et al., 2006). Tropospheric $O_3$ is a potent greenhouse gas and it also has
detrimental effects on human health and vegetation (Knowlton et al., 2004; Hollaway et al.,
2012; Yue and Unger, 2014). $NO_x$ is an important precursor for $O_3$ production and peroxyacetyl
nitrate (PAN), which serves as a reservoir for $NO_x$. PAN, $O_3$, as well as some of their precursors,
have relatively long lifetimes in the atmosphere, enabling them to be transported long distance to
remote regions such as the Arctic.
Recent studies have shown some significant challenges for atmospheric chemical transport
models to simulate $O_3$ and its precursors in the Arctic (e.g., Shindell et al., 2008; Alvarado et al.,
2010; Walker et al., 2012; Wespes et al., 2012; Fischer et al., 2014; Monks et al., 2015). In the
multi-model assessment by Shindell et al. (2008), more than a dozen models all showed
systematic and persistent underestimation of $O_3$ at the GEOSummit station, Greenland (hereafter
referred to as Summit). Alvarado et al. (2010) used $NO_x$ and PAN measurements from ARCTAS
(Arctic Research of the Composition of the Troposphere from Aircraft and Satellites) in the
summer to compare with model simulations. They found that model simulated $NO_x$ mixing ratios
were higher than observations, while PAN mixing ratios were lower in fresh boreal fire plumes.
In terms of global PAN simulations, Fischer et al. (2014) directly partitioned 40% of $NO_x$
emissions from wildfires to PAN formation, which improved the agreement between model and





observations. However, the model still underestimated PAN surface mixing ratios during springtime in the Arctic. Walker et al. (2012) reported that model simulated $O_3$ mixing ratios were biased low when compared with balloon data during summertime from two high-latitude sites at Eureka (80°N, 86°W) and Ny-Ålesund (79°, 12°E). Wespes et al. (2012) also revealed that model simulated $O_3$ mixing ratios below the boundary layer in the Arctic are significantly underestimated during spring-summer, compared with ARCTAS measurements. More recently, Monks et al. (2015) further demonstrated that model simulated $O_3$ mixing ratios in the Arctic at the surface and in the upper troposphere were generally lower than the observations.

Global anthropogenic ethane ($C_2H_6$) emission estimates range from 5.7 Tg/yr to 16.2 Tg/yr (Blake and Rowland, 1986; Kanakidou et al., 1991; Rudolph, 1995; Gutpa et al., 1998; Xiao et al., 2008; Etiope and Ciccioli, 2009; Pozzer et al., 2010; Aydin et al., 2011; Simpson et al., 2012; Emmons et al., 2015; Franco et al., 2016; Tzompa-Sosa et al., 2017), with a decreasing trend from 1980 to 2009 (Simpson et al., 2012; Helmig et al., 2014a). However, since 2009, global anthropogenic $C_2H_6$ emissions began to increase (Franco et al., 2015; Hausmann et al., 2016; Helmig et al., 2016). The RETRO (REanalysis of the TROpospheric chemical composition) global emission inventory used to be the global default anthropogenic $C_2H_6$ emission inventory, the annual budget of which has been shown too low compared with observations (Xiao et al., 2008; Fischer et al., 2014; Franco et al., 2015, 2016), whereas the emission inventory from Xiao et al. (2008) has been demonstrated to match observations during 1988-2004. Model simulated $C_2H_6$ mixing ratios are particularly biased low in the remote high latitude regions, when compared with observations (Emmons et al., 2015).

Field measurements at Summit show that snowpack emits gas-phase $NO_x$, PAN, nitrous acid (HONO), as well as hydrogen peroxide ($H_2O_2$) during spring-summer, when polar sun rises (Ford et al., 2002; Honrath et al., 2002). Although several 1-D models (Thomas et al., 2011, 2012; Frey et al., 2013; Murray et al., 2015) have validated its significant importance for surface $NO_x$ as well as $O_3$ formation, current global chemical transport models (CTMs) usually do not include this emission source (Zatko et al., 2016).

In this study, we employ a global chemical transport model, GEOS-Chem CTM, to evaluate the model performance for surface $O_3$ and its precursors over Summit, in conjunction with two years in-situ measurements during 2008-2010. This paper is organized as follows: section 2 describes



model methods and observations, followed by detailed comparisons of model simulations against
observations for $O_3$ and $O_3$ precursors in section 3; conclusions are summarized in section 4.

## 2. Observational data and model simulations

In situ measurements of $NO_x$, PAN, and non-methane hydrocarbons (NMHCs) were performed
at Summit from July 2008 to June 2010 (Helmig et al., 2014b; Kramer et al., 2015). An
automated $O_3$ chemiluminescence detection system was used to measure $NO_x$ (Ridley and
Grahek, 1990); a commercial PAN gas chromatography analyzer (PAN-GC, Metcon, In.,
Boulder, CO) was employed for the measurement of PAN. Measurements of NMHC relied on an
automated GC-Flame Ionization Detection (FID) system. Readers are referred to Kramer et al.
(2015) and Helmig et al. (2014b) for the details of the measurement techniques and equipment
setup. Surface measurements of $O_3$ using ultraviolet light absorption at 254 nm
(Petropavlovskikh and Oltmans, 2012), and CO by GC (Novellie and Masarie, 2015) are from
the National Oceanic and Atmospheric Administration (NOAA). Hourly averaged $O_3$ and flask
sampled CO between July 2008 and June 2010 were downloaded from the NOAA Earth System
Research Laboratory (ESRL) Global Monitoring Division (GMD) website
(http://www.esrl.noaa.gov/gmd/dv/data/). Vertical ozonesonde data profiles were also
downloaded from NOAA ESRL GMD (McClure-Begley et al., 2014).
The GEOS-Chem CTM (Bey et al., 2001) was used to simulate the seasonal cycles of $O_3$ and
related species ($NO_x$, PAN, NMHCs) at Summit. The GEOS-Chem model has fully coupled $O_3$-
$NO_x$-VOC-Aerosol chemistry mechanism and is driven by assimilated meteorological data from
the Goddard Earth Observing System version 5 (GEOS-5) of the NASA Global Modeling
Assimilation Office. The GEOS-Chem model has been extensively evaluated and applied in a
wide range of applications (Martin et al., 2002; Park et al., 2004; Wu et al., 2007; Hudman et al.,
2009; Johnson et al., 2010; Huang et al., 2013; Kumar et al., 2013; Zhang et al., 2014; Hickman
et al., 2017). GEOS-Chem v10-1 with grid resolution of 4˚ latitude by 5˚ longitude, and 47
vertical layers was used for the model control simulation. Following McLinden et al. (2000), the
Linoz stratospheric $O_3$ chemistry scheme was used. The simulation was run from June 2007 to
June 2010 and the results from the last two years were used in the final analysis. Time series data
were archived with 3-hr temporal resolution at the Summit grid box.





Global anthropogenic emissions of $NO_x$, $SO_2$, $NH_3$, and CO in the model are based on the
Emission Database for Global Atmospheric Research (EDGAR) v4.2 inventory, which is
overwritten by regional emission inventories where applicable, such as the BRAVO inventory
for Mexico (Kuhns et al., 2005), the CAC over Canada, the EMEP emissions over Europe, the
Model Inter-comparison Study for Asia Phase III (MIX) emissions over Asia (Li et al., 2017),
and the US EPA NEI 2011 (NEI11) emission inventory (Simon et al., 2010). Soil $NO_x$ emission
scheme follows Hudman et al. (2012). Lightning $NO_x$ emissions are calculated per flash rate
based on GEOS-5 computed cloud-top heights (Price and Rind, 1992), which are determined by
deep convection and constrained by satellite observations for monthly average flash rates from
the Lightning Imaging Sensor and Optical Transient Detector (OTD/LIS) (Sauvage et al., 2007;
Murray et al., 2012). Biomass burning emissions are from the Global Fire Emission Database
version 4 (GFED4) inventory with monthly resolution (Giglio et al., 2013). The RETRO  global
anthropogenic NMHC emission inventory (van het Bolscher et al., 2008) was used except for
$C_2H_6$ and propane ($C_3H_8$), which follows Xiao et al. (2008, hereafter referred to as X08) for the
year 2001. Global biofuel emission inventory follows Yevich and Logan (2003), which includes
emissions for $C_2H_6$ and $C_3H_8$. For biogenic VOC emissions, the Model of Emissions of Gases
and Aerosols from Nature (MEGAN) scheme (Guenther et al., 2006) was used. Dry deposition
of species in GEOS-Chem uses a standard resistance-in-series scheme (Wesely, 1989), as
implemented in Wang et al. (1998). Wet scavenging follows Liu et al. (2001), including
scavenging in convective updraft, rainout (in-cloud) and washout (below-cloud) from convective
anvils and large-scale precipitation.
We first ran the standard GEOS-Chem model with a-priori emissions and compare the
simulation results against observations for various species (including $NO_x$, PAN, $C_2H_6$, $C_3H_8$,
CO, and $O_3$, as shown in Fig. 1). Then we focus on the model-observation discrepancies, and
where applicable, make revisions to the model simulations and further evaluate the improvement
in model performance, as discussed in details below.
**3.  Results and Discussions**
3.1     $NO_x$



We first combine the two years of data for July 2008 – June 2010 and anylaze their seasonal
variations. As shown in Figure 1a, the GEOS-Chem model in general captures the abundance
and seasonal variation of $NO_x$ for July-October. However, compared to observations the model
results significantly (by a factor of 2) overestimate $NO_x$ mixing ratios for November-January,
while underestimating the data in spring and early summer by over 50%. Another challenge for
the model simulation is that it does not capture the decrease of $NO_x$ for May-December. We find
that during the 2009-2010 winter season, model simulations show several high $NO_x$ spikes with
peak $NO_x$ mixing ratios reaching ~ 0.15 ppbv or higher, which is ~ 15 times greater than typical
backgound levels (Fig. 2). These large peaks in $NO_x$ were not observed in the data. Similar peaks
were also seen in the model simulations during the 2008-2009 winter season; however, there are
no measurement data available for this period to compare with.
Further analyses show that the model-simulated high $NO_x$ spikes during wintertime are all
associated with transport events from Europe. We carried out a sensitivity study to examine the
impacts of European emissions on Arctic $NO_x$ by mannually reducing anthropogenic $NO_x$
emissions from the EMEP emission inventory over Europe by 50% (EMEP50). Results show
that surface peak $NO_x$ mixing ratios over Summit during the spike events (e.g., dates around
12/09/2009, 12/15/2009, 1/15/2010 and 1/22/2010) from EMEP50 almost decline proportionally
by approximately 50% during 2009/12/01-2010/01/31 (Fig. 2), which confirms  that the modeled
$NO_x$ spikes at Summit during wintertime are associated with transport from Europe. However,
the model simulated $NO_x$ is still significantly higher than observations. Comparisons for surface
$NO_2$ mixing ratios between model simulations and 11 in-situ observational sites over Europe
during this period were conducted with  data downloaded from http://ebas.nilu.no. For detailed
site information, $NO_2$ measurement technique and resolution, refer to Table 1. Measurment data
over these two months for each site were averaged to compare with the corresponding grid cell in
the model. As shown in Figure 3a, GEOS-Chem overestimates surface $NO_2$ mixing ratios at
these sites by over 66%, compared with observations.
Instead of using EMEP, we carried out another sensitivity study to force anthropogenic $NO_x$
emissions over Europe following EDGAR v4.2 (EURO_EDGAR), with other model
configurations identical to control simulations. As shown in Figure 2, the $NO_x$ mixing ratios over
Summit during 12/2009-01/2010 agree much better with observations, especially for January



2010 where the model captures the magnitudes of observational peaks (Fig. 2). This is because
$NO_x$ emissions from EDGAR over Europe (1.97 Tg NO) are 12% lower than that from EMEP
(2.24 Tg NO) for the months of 12/2009 and 01/2010. Furthermore, the discrepancy for the
differences of surface $NO_2$ mixing ratios over Europe between EURO_EDGAR and observations
is further reduced (by 50%), relative to the control runs (Fig. 3b). Similarly, we also tested the
sensitivty of surface $NO_x$ mixing ratios over Summit in response to the changes in the
anthropogenic $NO_x$ emissions from NEI11 over US and MIX over Asia (including Siberia)
during these two months, and found that surface $NO_x$ mixing ratio over Summit during these two
months were quite close to the control simulations (not shown), reflecting insensitivity to
emission perturbations from US and Asia. Therefore, we conclude that uncertainties in fossil fuel
$NO_x$ emissions of EMEP associated with transport events from Europe in the model are the most
likely cause for the wintertime $NO_x$ spikes over Summit.
For April-July, model simulated monthly mean $NO_x$ mixing ratios over Summit are a factor of
two lower than the observations (Fig. 4a). In-situ measurements at Summit by Honrath et al.
(1999, 2000a, 2000b, 2002) showed upward fluxes of $NO_x$ ($2.52 \times 10^8$ molecules $cm^{-2}$ $s^{-1}$) from
photolysis of nitrate in snowpack during the summertime, leading to enhancement in $NO_x$ levels
in the surface layer by approximately 20 pptv, which is comparable to surface $NO_x$ mixing ratios
in the Arctic from other sources. Similar results were found over the East Antarctic Plateau, a
remote Antarctic location (75.1° S, 123.3° E) covered by snow/ice sheet (Frey et al., 2013;
Legrand et al., 2014). The standard GEOS-Chem model does not include the photolysis of nitrate
from snowpack, implying a missing source for $NO_x$ in the Arctic/Antarctic boundary layer.
In order to test the sensitivity of model simulated surface $NO_x$ mixing ratios to the snowpack
emissions, we implement in the model a constant 24-hr $NO_x$ flux ~ $2.52 \times 10^8$ molecules $cm^{-2}$ $s^{-1}$
during April-July over Greenland (60-85° N, 20-60° W), following the measurements conducted
at Summit during summertime by Honrath et al. (2002). As a result, we find that on average, the
model simulated surface $NO_x$ mixing ratios for April to July over Summit are more than double
that from the control simulation, which improves the agreement between model and observations
(Fig. 4a). However, the model is still not able to reproduce the decreasing trends of $NO_x$ for
May-October. This may be caused by the seasonally decreasing $NO_x$ production rate in the



snowpack from spring to fall from a gradual depletion of the snowpack $NO_x$ reservoir (Van Dam
et al., 2015), whereas the model considers a simple constant $NO_x$ emission flux.

## 3.2    PAN

Figure 1b shows the comparison of model simulated monthly mean PAN mixing ratios with the
measurement data. The model captures the seasonal variation of PAN well, although
significantly (by ~30%) underestimting the PAN mixing ratios for April-June. By running the
model simulation with higher horizontal resolution at 2° latitude by 2.5° longitude (hereafter
referred to as GEOS-Chem 2x2.5), we find that the monthly mean PAN mixing ratios over
Summit during April-July are increased by up to 23.3 pptv compared to the 4x5 simulation (Fig.
4b). This can be explained by two reasons. First, coarse model resolution (e.g., 4x5 horizontal
resolution) could artificially smear the intense emission sources throughout the entire grid cell
(e.g., over urban regions), leading to underestimates of downwind concentrations for species,
e.g., $O_3$ and $O_3$ precursors (Jang et al., 1995; Yu et al., 2016).  Second, ventilation of lower
atmosphere could be better resolved by fine model resolution, leading to more efficient vertical
advection (Wang et al., 2004; Chen et al., 2009; Yu et al., 2016). However, on average, monthly
mean model simulated PAN mixing ratios are still underestimated by 20% during this period,
compared with observations.
Snowpack can emit not only $NO_x$, but also PAN, based on field studies at Summit during
summertime by Ford et al. (2002). GEOS-Chem does not contain snowpack PAN emissions and
chemistry. For a sensitivity study, similar to snowpack $NO_x$ emissions as discussed in section
3.1, we considered a 24-hr constant flux of $2.52 \times 10^8$ molecules $cm^{-2}$ $s^{-1}$ of PAN from April to
July, following Ford et al. (2002). As a result, model simulated PAN mixing ratios agree much
better with observations (Fig. 4b). Note that there are also other possible reasons that lead to
model bias. For instance, a study by Fischer et al. (2014) showed that an improved agreement
between modeled and measured PAN in the high latitudes can be found when the model was
forced to emit a portion of the fire emissions above the boundary layer as well as by directly
partitioning 40% of $NO_x$ emissions from fires into PAN. However, in our case, we did not find
much difference beteween a sensitivity study following this method and our control runs.

## 3.3    NMHC



Comparisons of observed surface $C_2H_6$ and $C_3H_8$ mixing ratios with GEOS-Chem simulations at Summit are shown in Figures 1c and d. The model simulations agree well with surface measurements of $C_3H_8$, but systematically overestimate $C_2H_6$ (by approximately 25% annually), with the largest bias (0.48 ppbv) occuring during summer. This is consistent with the study from Tzompa-Sosa et al., (2017), which used the same model as our study and pointed out that using X08 as global anthropogenic $C_2H_6$ emission inventory systematically overestimated surface $C_2H_6$ mixing ratios over the Northern Hemisphere, compared with ground-based observations. Anthropogenic $C_2H_6$ emissions over US from NEI11 are shown to geographically match the distribution of active oil and natural wells (Tzompa-Sosa et al., 2017), and the most recent MIX has been updated to synergize anthropogenic $C_2H_6$ emissions from various countries over Asia (Li et al., 2017). Therefore, instead of using global anthropogenic fossil fuel emissions of $C_2H_6$ following X08, we first conducted sensitivity simulations by overwritting global emission inventories by NEI11 over US, and MIX over Asia (hereafter referred to as NEI11_MIX). Both NEI11 and MIX contain emissions for the years from 2008 to 2010, which could realistically represent the annual and seasonal variations of $C_2H_6$ emissions over the US and Asia, thus spatially and temporally better representative of anthropogenic $C_2H_6$ emissions from mid-latitudes transported to the Arctic regions. In general, model control simulations overestimate annual mean surface $C_2H_6$ mixing ratios primarily in the Northern Hemisphere, with large differences occurring over Asia and US by up to 5 ppbv, compared with NEI11_MIX during the period of 07/2008-06/2010 (Fig. S1). All the above changes are driven by the substantial reductions of anthropogenic $C_2H_6$ emissions between emission inventories, from 3.5 (X08) to 2.5 Tg/yr (MIX) over Asia, and from 1.9 Tg/yr (X08) to 1.4 Tg/yr (NEI11) over US, reflecting decreasing trend of anthropogenic $C_2H_6$ emissions during 2001-2009 because X08 emission inventory is based on the year 2001, which is consistent with Helmig et al. (2014a). Substantial changes in surface $C_2H_6$ mixing ratios over the US bewteen control simulations and NEI11_MIX reflects that there exist tempospatial changes of $C_2H_6$ emissions from oil and gas productions during the period of 2001-2009. A similar pattern was also found by Tsompa-Sosa et al. (2017). In contrast to the control simulations, NEI11_MIX model simulations show that monthly mean $C_2H_6$ mixing ratios over Summit are systematically underestimated by 24%, compared with observations (Fig. 5). Tsompa-Sosa et al. (2017) reported that NEI11 for $C_2H_6$ emissions were likely underestimated by 40%, compared with in-situ and aircraft observations over the US. With



NEI11 $C_2H_6$ emissions increases by 40%, however, model simulated annual mean $C_2H_6$ mixing
ratios over Summit only increase by 6% during the period of 07/2008-06/2010, relative to
NEI11_MIX.
Similar to NEI11_MIX, we further conducted sensitivity studies by only replacing the regional
emission inventory for $C_2H_6$ over the US, with other regions still following X08 (hereafter
referred to as NEI11_ONLY). Consequently, model simulated surface $C_2H_6$ mixing ratios over
Summit agree better with observations during winter-spring (Fig. 5), decreasing the bias from
+15% (control simulations) to +6%. However, model simulated $C_2H_6$ mixing ratios during
summer-fall are higher than the observations by over 30%.
We then scaled up the MIX emissions for $C_2H_6$ by 20% over Asia, with other model
configurations identical to NEI11_MIX (hereafter referred to as NEI11_MIX20). By doing this,
we increase fossil fuel $C_2H_6$ emissions from 2.5 to 3 Tg/yr. We find that annual mean $C_2H_6$
mixing ratios from NEI11_MIX20 agree quite well with observations over Summit, with bias
less than 1% (Fig. 5). This implies that further assessments of anthropogenic $C_2H_6$ emissions
from MIX over Asia are needed and a more accurate global anthropogenic $C_2H_6$ emission
inventory should be developed and validated to replace X08 in the future. Note that this standard
version of GEOS-Chem does not account for the sink of $C_2H_6$ from the reaction with chlorine,
which could reduce the global annual mean surface $C_2H_6$ mixing ratios by 0-30%, and the global
burden of $C_2H_6$ by about 20%, compared with the simulation without considering the halogen
chemistry (Sherwen et al., 2016), which introduces additional uncertainty for our measurement-
model comparison, together with the highly uncertain seasonality of $C_2H_6$ chemistry.
**3.4 CO**
Figure 1e shows the comparison of model simulated CO mixing ratios with observations over
Summit. Overall, the model generally captures the seasoanl trend and annual mean of CO, with
annual mean model simulated CO mixing ratios slightly overestimated by up to 3 ppbv,
compared with observations.
**3.5 O₃**





Surface $O_3$ mixing ratios from model simulations and surface observations are compared in
Figure 1f. The GEOS-Chem model captures the seasonal variation of $O_3$ including the spring
peak. However, the model shows a systematic low bias for most time of the year, in particualr for
April–July when the surface $O_3$ mixing ratios are underestimated by approximately 13% (~ 6.5
ppbv). Here we focus our analysis for the possible causes that lead to the model low bias during
April-July.
As discussed earlier, snowpack emissions due to the photolysis of nitrate in the snow during late
spring and summer could contribute to $NO_x$ and HONO levels in the ambient air which could
significantly enhance $O_3$ production (Crawford et al., 2001; Zhou et al., 2001; Dibb et al., 2002;
Honrath et al., 2002; Yang et al., 2002; Grannas et al., 2007; Helmig et al., 2008; Legrand et al.,
2014). We ran a sensitivity study to test the response of surface $O_3$ mixing ratios to the
perturbations of $NO_x$ and HONO from snowpack emissions. In addtion to snowpack $NO_x$
emissions that are described in Section 3.1, we implemented in the model a constant flux of
HONO ($4.64 \times 10^7$ molecules $cm^{-2}$ $s^{-1}$) from April to July over Summit as well (Honrath et al.,
2002). As a result, monthly mean model simulated surface $O_3$ mixing ratios increase by up to 3
ppbv during this period (Fig. 6). The largest effect occurs in July due to relatively strong solar
radiation. $O_3$ formation due to snowpack emissions in our study is slightly higher than that in
Zatko et al. (2016) because HONO from snowpack emissions is not considered in their study.
However, for the months of April and May, surface $O_3$ mixing ratios only increase by ~ 1 ppbv,
compared with the control runs. That is, even after accounting for the snowpack emissions, the
model simulated $O_3$ mixing ratios are still significantly lower than the observations.
Comparison of the model simulations with different resolutions (4x5 vs. 2x2.5) shows that the
finer resolution simulations substantially increase monthly mean $O_3$ mixing ratios over Summit
by up to 6 ppbv for the months of June and July (Fig. 6). As discussed in section 3.2, fine model
resolution can better resolve the emission strengths, which could significantly affect downwind
chemical reactions, e.g., $O_3$ production efficiency (Liang and Jacobson, 2000). Moreover, terrain
elevations from fine model resolution are better represented (thus better representative of
Summit's elevation) and more efficient vertical ventilation of $O_3$ and $O_3$ precursors can be
achieved (Wang et al., 2004). Together with the impact of snowpack chemistry, this brings
model simulated surface $O_3$ mixing ratios over Summit in much better agreement with



observations for these two months. Unfortunately, there is still a low bias in the model for the
months of April and May.
Another possible cause for the $O_3$ biases between model simulations and observations is the
stratosphere-to-troposphere exchange scheme (STE) in the model for $O_3$. Liang et al. (2011)
have pointed out that STE could be a significant direct sources of $O_3$ in the Arctic during spring-
summer. We retrieved vertical profiles of $O_3$ mixing ratios and specific humidity from
ozonesondes (0-5 km elevation above the Summit surface) launched at Summit for the months of
June and July in 2008 and compared those data with model control runs. Ozonesondes were
launched intensively during these two months (a total of 19 times). As shown in Figure 7,
compared with observations, model simulated $O_3$ mixing ratios averaged over 0-5 km above the
ground level are underestimated by 3% and 9% in June and July 2008 (Fig. 7a). However,
specific humidity in GEOS-5 is overestimated by 50% and 81% (Fig. 7b) respectively.
Ozonesonde data show that Summit frequently encounters high $O_3$/low water vapors events (e.g.,
July 9-11, 2008), which are likely of upper tropospheric/stratospheric origins (Helmig et al.,
2007b), but these are not captured by the model, which implies that GEOS-Chem possibly
underestimates STE for $O_3$ over Summit.
Boundary layer height is another factor that could potentially affect the discrepancy of $O_3$ mixing
ratios between model and observations. The mean springtime afternoon (12:00-14:00, local time)
boundary layer height in the model at Summit for the year 2009 is 160 m, which agrees fairly
well with observations (156 m) at Summit conducted in spring 2005 (Cohen et al., 2007).
Therefore, we exclude that model uncertainties in boundary layer height representation in
springtime cause the low bias of $O_3$ mixing ratios between model and observations.
Lastly, we also compared $O_3$ dry deposition velocity ($V_{dry}$) in the model in springtime with
observations at Summit. For spring 2009, mean $O_3$ $V_{dry}$ in the model at Summit is 0.009 cm s$^{-1}$,
which is within the range of -0.01-0.01 cm s$^{-1}$ observed at Summit (Helmig et al., 2009). Helmig
et al. (2009) revealed that afternoon (12:00-18:00, local time) $O_3$ $V_{dry}$ during springtime was
close to 0.01 cm s$^{-1}$. For other times during the day, $O_3$ $V_{dry}$ was either close to zero or negative
(i.e., $O_3$ production over the snow outweighs its dry deposition). As a result, the net mean $O_3$ $V_{dry}$
from observations during springtime is about 4-6 times lower than model simulations. Therefore,





model overestimation of $O_3$ loss via surface uptake is another factor that contributes to the low
bias of surface $O_3$ mixing ratios at Summit in the model during springtime.

## 4. Conclusions

We combine model simulations with two-year (July 2008-June 2010) ground based
measurements at Summit to better understand the abundance and seasonal variations of
tropospheric $O_3$ and related species in the Arctic. In general, the GEOS-Chem model is capable
of reproducing the seasonal cycles of $NO_x$, PAN, $C_2H_6$, $C_3H_8$, CO, and $O_3$. However, some major
discrepancies between model and observations, especially for $NO_x$, PAN, $C_2H_6$, and $O_3$ are also
identified.
There are significant differences between model simulated $NO_x$ mixing ratios and observations
for the spring and winter seasons. The model underestimates $NO_x$ mixing ratios by
approximately 50% during late spring to early summer, which is likely due to the missing $NO_x$
emissions from nitrate photolysis in the snowpack. At the same time, the model overestimates
$NO_x$ mixing ratios by more than a factor of two in wintertime. Model simulations indicate
episodic but frequent transport events from Europe in wintertime leading to $NO_x$ spikes reaching
15 times typical $NO_x$ mixing ratios at Summit; these large $NO_x$ spikes are not seen in the
observational data. We have carried out multiple sensitivity model studies but are still unable to
reconcile this discrepancy.
The model successfully captures the seasonal cycles and the spring maximum PAN mixing
ratios, although it underestimates PAN by over 30% during late spring and early summer. Model
sensitivity studies reveal that this discrepancy could be largely resolved by accounting for PAN
emissions from snowpack.
For $C_3H_8$ and CO, model simulations overall agree well with the surface measurements,
however, the model tends to systematically overestimate surface $C_2H_6$ mixing ratios by
approximately 20% on an annual average, compared with observations. This may be explained
by that annual emission budgets of $C_2H_6$ over US and Asia from X08 emission inventory are
higher than those from NEI11 and MIX by over 40%. By replacing X08 over US with NEI11 for
$C_2H_6$, and scaling up MIX by 20%, the model-observation bias can be resolved, with annual
mean bias less than 1%. However, care must be taken to interpret this result because we do not




take into account other factors that may influence the discrepancy of surface $C_2H_6$ mixing ratios
at Summit between model and observations, such as the $C_2H_6$ chemistry with cholorine.
GEOS-Chem is able to reproduce the seasonal variation of surface $O_3$ at Summit but persistently
underestimates $O_3$ mixing ratios by approximately 13% (~ 6.5 ppbv) from April to July. This low
bias is likely caused by a combination of misrepresentations, including the missing snowpack
emissions of $NO_x$ and HONO, inaccurate representation of Summit's elevation from coarser
model resolutions, model overestimated $O_3$ dry deposition velocity during springtime, as well as
the underestimated STE.
All the results presented above reveal the importance of local snowpack emissions in regulating
the air quality over Arctic. Improvements in global CTM could likely be achieved by coupling
snowpack emissions of reactive gases and photochemistry modules in order to better simulate $O_3$
and $O_3$ precursors over snow and ice in the Arctic (Zatko et al., 2016). Moreover, this study also
demonstrates that anthropogenic emissions from midlatitudes play an important role in affecting
the Arctic air quality. However, further investigations in anthropogenic $NO_x$ emissions over
Europe and $C_2H_6$ emissions over Asia and North America are needed. The uncertainties in $O_3$
dry deposition and STE scheme in GEOS-Chem are warranted to be better quantified in our
future study.
**Acknowledgements** This research was funded by U.S. EPA grant  83518901). Its contents are
solely the responsibility of the grantee and do not necessarily represent the official views of the
U.S. EPA. Further, U.S. EPA does not endorse the purchase of any commercial products or
services mentioned in the publication. Superior, a high performance computing cluster at
Michigan Technological University, was used in obtaining results presented in this publication.
L. J. Kramer, D. Helmig and R. E. Honrath thank NASA (grant NNX07AR26G) for supporting
the measurements at Summit. We acknowledge the observational dataset of $O_3$ and CO provided
by NOAA ESRL. Technical supports from M. Sulprizio and C. Keller are also acknowledged.






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



**Table 1.** Surface NO₂ measurements over Europe during 2009/12/01-2010/01/31.

| Site ID | Site name | Lat. (°N) | Lon.(°E) | Altitude a.s.l (m) | Technique | Resolution |
|---|---|---|---|---|---|---|
| BE0001R | Offagne | 49.88 | 5.20 | 430 | chemiluminescence | hourly |
| BE0032R | Eupen | 50.63 | 6 | 295 | chemiluminescence | hourly |
| DE0001R | Westerland | 54.93 | 8.31 | 12 | NaJ_solution | daily |
| DK0008R | Anholt | 56.72 | 11.52 | 40 | UV_fluorescence | hourly |
| FI0096G | Pallas | 67.97 | 24.12 | 340 | chemiluminescence | hourly |
| GB0014R | High Muffles | 54.33 | -0.8 | 267 | chemiluminescence | daily |
| NL0009R | Kollumerwaard | 53.33 | 6.28 | 1 | chemiluminescence | hourly |
| NO0001R | Birkenes | 58.38 | 8.25 | 190 | glass sinter | daily |
| NO0039R | Kårvatn | 62.78 | 8.88 | 210 | glass sinter | daily |
| NO0056R | Hurdal | 60.37 | 11.08 | 300 | glass sinter | daily |
| SE0005R | Bredkälen | 63.85 | 15.3 | 404 | abs_tube | daily |















**Fig. 1.** Box plot comparison for seasonal variations of (a) NOₓ, (b) PAN, (c) C₂H₆, (d) C₃H₈, (e)
CO, and (f) O₃ between GEOS-Chem model simulations (red) and in-situ measurements (blue)
over Summit for the period of 2008/07-2010/06. Data shown are monthly averages during this
period. The thick (thin) bars represent the 67% (95%) confidence intervals. Black and green dots
represent median and mean values, respectively. The statistics are based on daily averages.




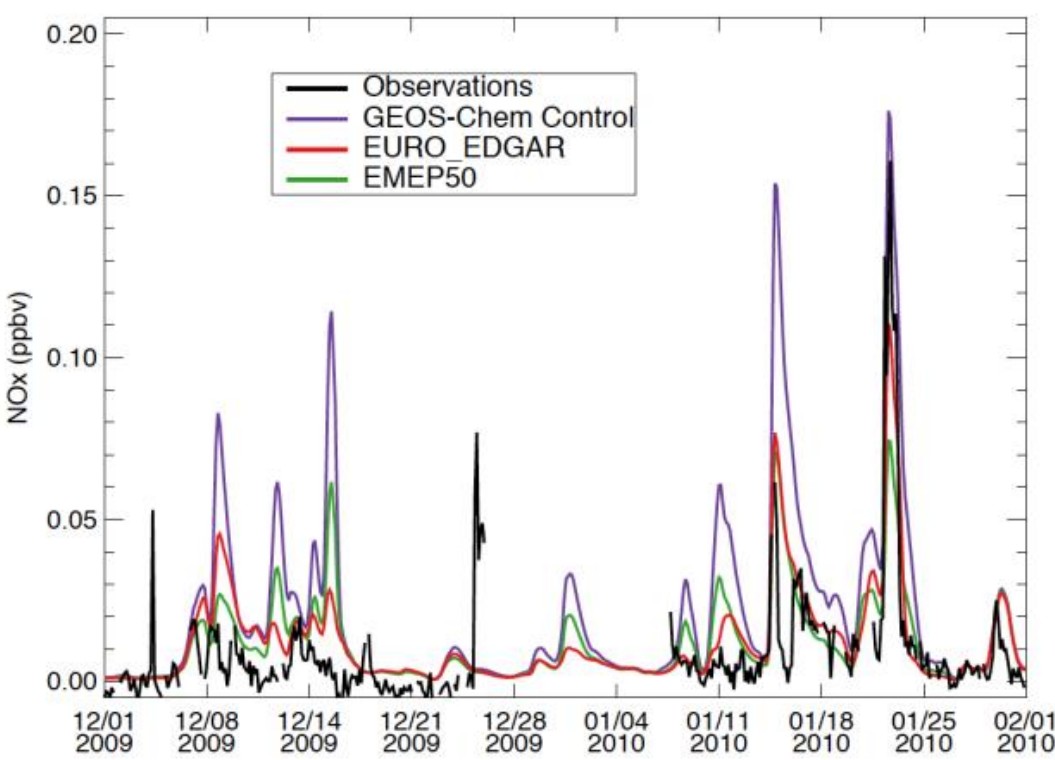


**Figure 2.** Timeseries of surface NOx mixing ratios over Summit from observations, GEOS-Chem model control simulations, EURO_EDGAR, and EMEP50 during 2009/12/01-2010/01/31.













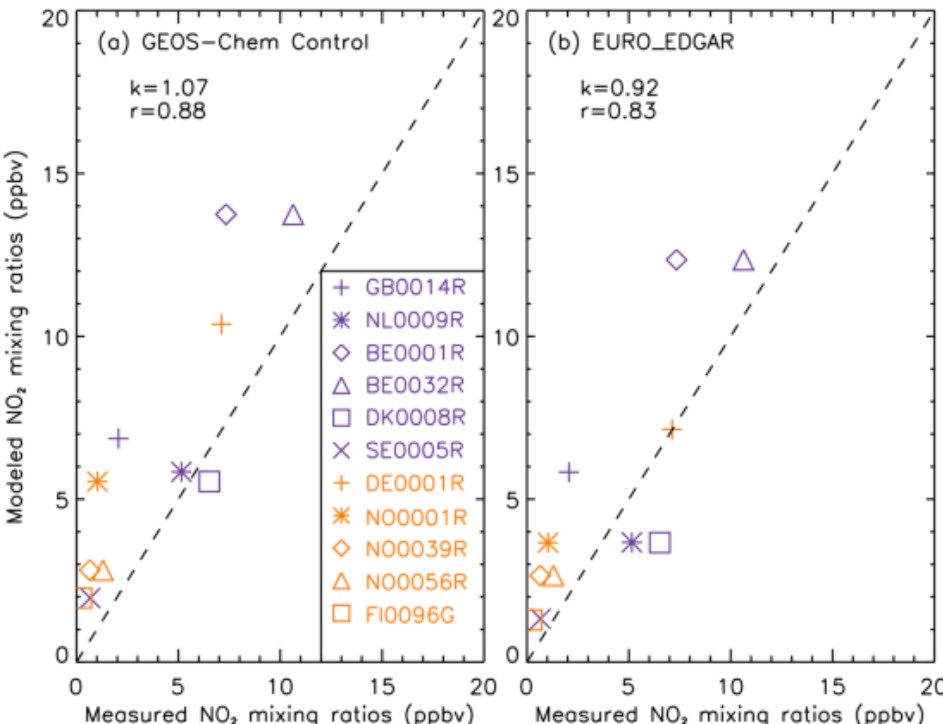


**Figure 3.** Scatter plots between measured monthly mean $NO_2$ mixing ratios at 11 observational

sites over Europe and model simulations from (a) GEOS-Chem control simulations and (b)

EURO_EDGAR during 2009/12/01-2010/01/31; also shown is the corresponding model-to-

observation slopes (k) and correlation coefficients (r) for each panel. The dash line is the 1:1

ratio. Explanations of site abbreviations are listed in Table 1.







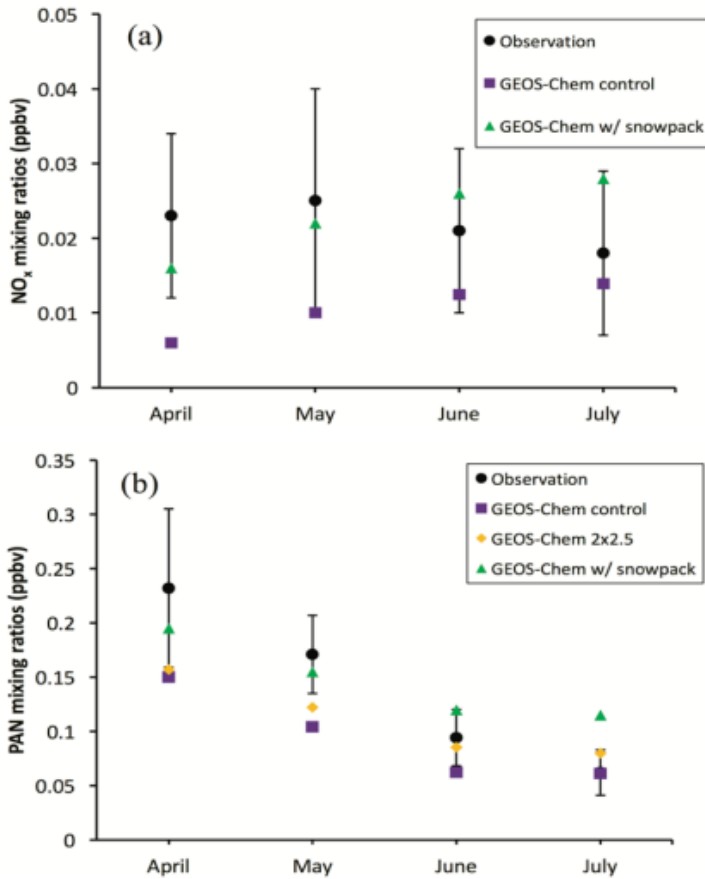


**Figure 4.** Monthly mean surface (a) NO$_x$ and (b) PAN mixing ratios from observations (black

circles), simulations with (green triangles) /without (purple squares) snowpack emissions, and

GEOS-Chem simulations with horizontal grid resolution 2° x 2.5° (orange diamonds) over the

period of April- July during 07/2008-06/2010. Vertical bars denote standard deviations over the

course of observations for each month.



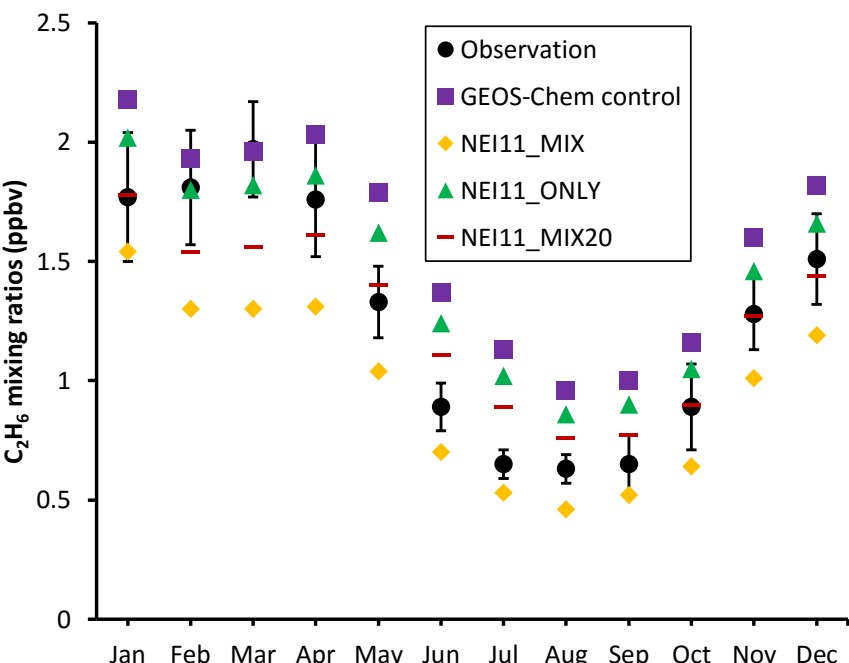

794

**Figure 5**. Monthly mean surface $C_2H_6$ mixing ratios at Summit from observations (black
circles), GEOS-Chem model control simulations (purple squares), NEI11_MIX (orange
diamond), and NEI11_ONLY (green triangles) simulations during 2008-2010; vertical bars
denote the standard deviation over the course of observations for each month.













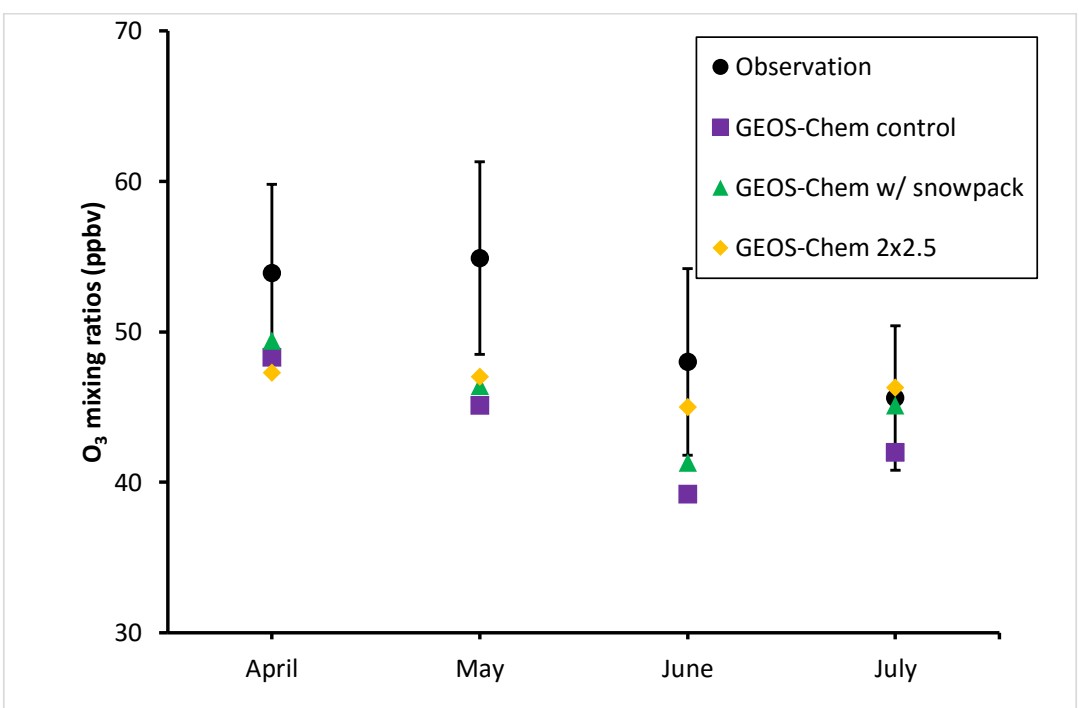


**Figure 6**. Monthly mean surface $O_3$ mixing ratios from observations (black circles), GEOS-Chem control runs (purple squares), with snowpack chemistry (green triangles), and horizontal grid resolution 2° x 2.5° (orange diamonds) for April-July. Vertical bars denote the variability over the course of observations for each month.













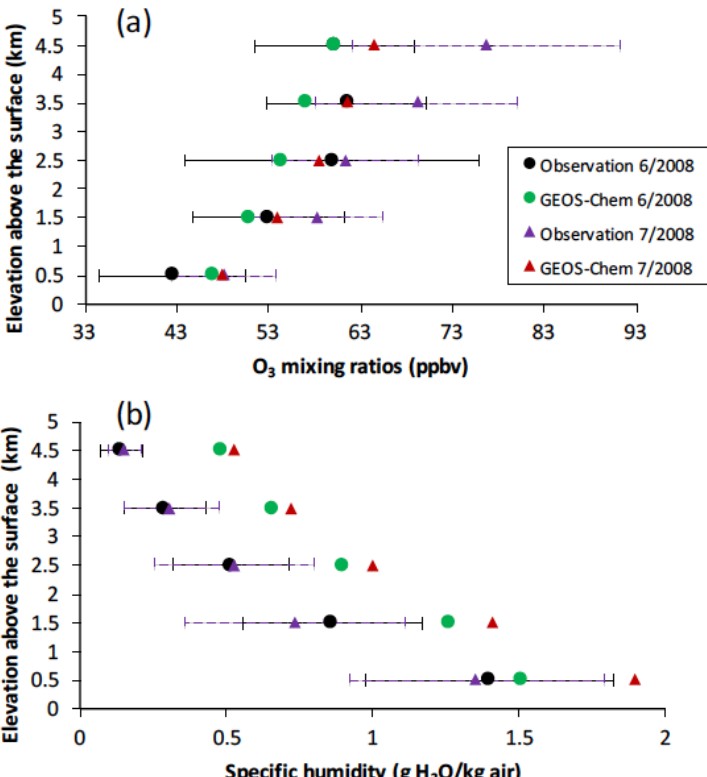


**Figure 7.** Comparisons of vertical profiles of (a) $O_3$ and (b) specific humidity between GEOS-Chem simulations and ozonesondes in June and July 2008 respectively, averaged over 1-km altitude bins. Black and green solid circles represent observations and simulations in June 2008 while purple and red triangles denote observations and simulations for July 2008 respectively. Solid and dash horizontal error bars represent observational standard deviations for June and July respectively.

827