# Peer review of "Surface ozone and its precursors at Summit, Greenland: comparison between observations and model simulations Yaoxian Huang1,a, Shiliang Wu1,2,3, Louisa J. Kramer1,2,b, Detley Helmig4, and Richard E. 3 4 Honrath1,2,† 5 1Department"

_Atmospheric Chemistry and Physics, 2017_

## Referee Comment (RC1) · H. Liu (Referee) · 9 Aug 2017

This paper presents a GEOS-Chem model analysis of surface ozone and its precursors (NOx, PAN, C2H6, C3H8, CO) observed at Summit, Greenland during the period of July 2008 - June 2010, with a focus on their concentrations and seasonal variations. Modeling tropospheric ozone in the Arctic has been challenging, and it is very interesting to use a state-of-the art chemical transport model to test and improve our understanding of its sources and variability. The authors identified the discrepancies between the GEOS-Chem simulations and observations, which were then examined using various model perturbation experiments. The results are original, and the paper

is concise and very well written. I recommend its publication on ACP with some minor modifications, as itemized below.

1). Title - Using "tropospheric ozone" in the title is a bit misleading. Although this study also compared the model vertical profiles of ozone and specific humidity with ozonesonde observations, the main scope of this paper is "surface ozone".

2). Section 2: It is not clear which version of the GEOS-5 meteorological data archive was used. Is it GEOS-5.1.0 or GEOS-5.2.0? See e.g., "http://wiki.seas.harvard.edu/geos-chem/index.php/GEOS-5_met_field_reprocessing" and "http://wiki.seas.harvard.edu/geos-chem/index.php/List_of_GEOS-5_met_fields".

3). Section 2, 2nd paragraph: "Time series data were archived with 3-hr temporal resolution at the Summit grid box" — I think you meant "grid column". Moreover, it is not clear how the model output was sampled in the vertical. The elevation of Summit is 3212m asl. Did you sample the model bottom layer, or the model vertical layer that is about 3212m above the sea level? The latter may very well be different than the former because the topography is not well resolved at coarse resolution. Would the results about model overestimates or underestimates found in this paper be different if the alternative way of model sampling is used (e.g., lines 206-207)?

Minor editorial comments:

Line 43: change the "and" before "volatile organic compounds" to comma.

Line 56: the ARCTAS mission

Line 66: What do you mean "O3 mixing ratios below the boundary layer"? Within the boundary layer?

Line 77: "....used to be the global default anthropogenic C2H6 emission inventory" - Do you mean "default" in GEOS-Chem or any other models?

Line 228: add "over Greenland" after "PAN".

Line2 268-270: "relative to NEI11_MIX" – isn't this relative to NEI11? Remove it?

Line2 279-281: The annual mean agrees quite well with observations, but the simulation is worse in summer.

Lines 283-288: Good point, but this long sentence needs a break.

Line 325: "Unfortunately, ..." – "However, ..."

Lines 339-340: "..., which implies that GEOS-Chem possibly underestimates STE for O3 over Summit" – This is interesting and appears consistent with Choi et al., ACP 2017 (https://www.atmos-chem-phys.net/17/8429/2017/ , see their Fig. 6) , where the GMI CTM driven by MERRA (GEOS-5.2.0) underestimates ozonesonde-observed ozone in the Northern Hemisphere high-latitude upper troposphere.

Lines 358-359: Summit, Greenland; surface ozone

Figures 2, 3,5, S1: In the caption, state briefly what the perturbation simulations are and refer the reader to the text for details.

---

## Referee Comment (RC2) · Anonymous Referee #1 · 31 Aug 2017

This paper describes an evaluation of tropospheric ozone and its precursor species simulated by the GEOS-Chem global chemical transport model (CTM) at the Summit observatory station in Greenland. Based on evaluation of the standard GEOS-Chem model, and deficiencies identified through comparison with observations, a number of model changes are implemented (mostly to emissions) which are shown to improve the model performance. The paper serves as a useful documentation of Greenland surface ozone, NOy and VOC sensitivity to a number of key processes, and highlights processes that warrant further investigation to improve understanding of the surface Arctic ozone budget. These issues are important in light of recent studies demonstrating poor model performance for Arctic tropospheric ozone, as cited by the authors.

[Figure]

The paper is generally well written, logically structured and is suitable for the journal. I would recommend publication of this manuscript in ACP, once the following minor issues have been addressed.

1) Paragraph beginning Line 69. The discussion of ethane appears a bit out of the blue. The authors should explain more clearly in the manuscript the importance and relevance of ethane to the previous discussion. i.e. give some context for how ethane is relevant to the study - which is motivated by understanding Arctic tropospheric ozone. i.e. as has been done for NOx, PAN.

2) Lines 109-112: It is unclear here what is meant by fully coupled aerosol? Does this include size-resolved modal aerosol for example? Heterogeneous chemistry, semi-volatile nitrate..?

3) Lines 112-115: Discussion of previous GEOS-Chem evaluation. It would be helpful here to provide a few sentences for a brief but more critical review of what has been shown in terms of model performance with previous studies specifically using GEOS-Chem in the Arctic. e.g. sensitivity analysis by Christian et al., (2107), the recent POLMIP evaluation (see Monks et al., 2015). These have shown some important limitations and strengths that it would be useful to point out for context.

4) Discussion of model NOx bias (first paragraph of page 5). Perhaps here quote the obs/model slope or model bias. You give figures for the slopes / correlations in the panels of Fig 3 but don't mention the numbers in the text.

5) Lines 206-208: Is the magnitude of the snowpack NOx reservoir depletion of right order to explain this? Is the source linearly dependent on the reservoir? Would it be hard to test this in the model to see if it improves the model bias? i.e. can you scale the monthly emissions according to this finding? Perhaps not necessary, but a brief discussion of the order of magnitude of depletion and how that relates to the model bias would be helpful.

6) Does this model include the PAN budget updates from the Fischer et al. study that is mentioned? This should be stated clearly. Arnold et al., (2105) showed that GEOS-Chem produces less PAN relative to CO than other models in Arctic air masses influenced by fires. It would be useful to refer back to this here to give context to the model performance relative to that found for other models.

Typographical / editorial corrections:

Line 58: "...while PAN mixing ratios were lower in fresh boreal fire plumes." This sentence in unclear. Lower than observed? Lower than in other air mass types simulated in the model?

Line 82: ".. that the snowpack emits.."

Paragraph beginning Line 141 contains mixed (past / present) tenses. Please adjust the text to make it consistent.

Line 156: " not observed in the data." Better to simply say ".. not observed".

Line 161: Omit word "mannually" (which should be spelled "manually" in any case).

References

Arnold, S. R., Emmons, L. K., Monks, S. A., Law, K. S., Ridley, D. A., Turquety, S., Tilmes, S., Thomas, J. L., Bouarar, I., Flemming, J., Huijnen, V., Mao, J., Duncan, B. N., Steenrod, S., Yoshida, Y., Langner, J., and Long, Y.: Biomass burning influence on high-latitude tropospheric ozone and reactive nitrogen in summer 2008: a multi-model analysis based on POLMIP simulations, Atmos. Chem. Phys., 15, 6047-6068, https://doi.org/10.5194/acp-15-6047-2015, 2015.

Christian, K. E., Brune, W. H., and Mao, J.: Global sensitivity analysis of the GEOS-Chem chemical transport model: ozone and hydrogen oxides during ARCTAS (2008), Atmos. Chem. Phys., 17, 3769-3784, https://doi.org/10.5194/acp-17-3769-2017, 2017.

Monks, S. A., et al., Multi-model study of chemical and physical controls on transport of anthropogenic and biomass burning pollution to the Arctic, Atmos. Chem. Phys., 15, 3575- 3603, doi:10.5194/acp-15-3575-2015, 2015.
* * *

---

## Author Comment (AC1) · 13 Oct 2017

Response to reviewer #1:

We thank Reviewer #1 for his/her valuable and thoughtful comments. Our responses to the comments are provided below, with the reviewer's comments italicized and our responses in plain and bold fonts.

*This paper describes an evaluation of tropospheric ozone and its precursor species simulated by the GEOS-Chem global chemical transport model (CTM) at the Summit observatory station in Greenland. Based on evaluation of the standard GEOS-Chem model, and deficiencies identified through comparison with observations, a number of model changes are implemented (mostly to emissions) which are shown to improve the model performance. The paper serves as a useful documentation of Greenland surface ozone, NO$_y$ and VOC sensitivity to a number of key processes, and highlights processes that warrant further investigation to improve understanding of the surface Arctic ozone budget. These issues are important in light of recent studies demonstrating poor model performance for Arctic tropospheric ozone, as cited by the authors. The paper is generally well written, logically structured and is suitable for the journal. I would recommend publication of this manuscript in ACP, once the following minor issues have been addressed.*

*1) Paragraph beginning Line 69. The discussion of ethane appears a bit out of the blue. The authors should explain more clearly in the manuscript the importance and relevance of ethane to the previous discussion. i.e. give some context for how ethane is relevant to the study - which is motivated by understanding Arctic tropospheric ozone. i.e. as has been done for NOx, PAN.*

**Response: We thank the reviewer for pointing this out. We have reorganized the flow of the text related to ethane in the introduction section -  a) we have deleted some discussions on ethane that are not closely related to our study here; b) We have added discussions on the importance of volatile organic compounds (e.g., ethane and propane) for the productions of ozone (lines 43-48) –**
**"Tropospheric ozone (O$_3$) and its precursors, including nitrogen oxides (NO$_x$ = NO + NO$_2$), carbon monoxide (CO), and volatile organic compounds (VOCs, such as ethane, propane, etc.) are important atmospheric species affecting both air quality and climate (e.g., Jacob et**

al., 1992; Fiore et al., 2002; Unger et al., 2006; Hollaway et al., 2012). Tropospheric $O_3$ is mainly produced by photochemical oxidation of CO and VOCs in the presence of $NO_x$, with additional contribution by transport from the stratosphere."

*2) Lines 109-112: It is unclear here what is meant by fully coupled aerosol? Does this include size-resolved modal aerosol for example? Heterogeneous chemistry, semivolatile nitrate..?*

**Response: We have clarified this part to "Simulations of $O_3$ and related species ($NO_x$, PAN, NMHCs) are conducted using the GEOS-Chem model (Bey et al., 2001) with coupled $O_3$-$NO_x$-VOC-Aerosol chemistry mechanism (i.e. these species interact with each other in the model)." (lines 103-105)**

*3) Lines 112-115: Discussion of previous GEOS-Chem evaluation. It would be helpful here to provide a few sentences for a brief but more critical review of what has been shown in terms of model performance with previous studies specifically using GEOS-Chem in the Arctic. e.g. sensitivity analysis by Christian et al., (2107), the recent POLMIP evaluation (see Monks et al., 2015). These have shown some important limitations and strengths that it would be useful to point out for context.*

**Response: This is an excellent point. We have added the descriptions of previous GEOS-Chem evaluations in the text. In the Introduction part, we have included the discussions of Monks et al. (2015) and Christian et al. (2017) as "More recently, Monks et al. (2015) further demonstrated that model simulated $O_3$ mixing ratios in the Arctic at the surface and in the upper troposphere were generally lower than the observations. In addition, a recent study by Christian et al. (2017) compared $O_3$ observations from the ARCTAS campaign to GEOS-Chem model simulations and found consistent low biases with the model simulated $O_3$ at all altitudes except the surface." (lines 70-74)**
**In Section 2, we have modified text in lines 112-115 in ACPD as "The GEOS-Chem model has been extensively evaluated and applied in a wide range of applications (Martin et al., 2002; Park et al., 2004; Wu et al., 2007; Hudman et al., 2009; Johnson et al., 2010; Huang et al., 2013; Kumar et al., 2013; Zhang et al., 2014; Hickman et al., 2017), including the**

**studies in the Arctic (e.g., Alvarado et al., 2010; Monks et al., 2015; Christian et al., 2017).”** **(lines 107-111)**

*4) Discussion of model NOx bias (first paragraph of page 5). Perhaps here quote the obs/model slope or model bias. You give figures for the slopes / correlations in the panels of Fig 3 but don't mention the numbers in the text.*

**Response: We agree with the reviewer. Now we have included model NO$_x$ bias in the text as “As shown in Figure 1a, the GEOS-Chem model simulated NO$_x$ agree well with the observations for July-October. However, compared to observations, the model results significantly overestimate NO$_x$ mixing ratios for November-January by about 150%, while underestimating the data in spring and early summer by approximately 60%.” (lines 147-150)**

**For Fig. 3, we have included the NO$_x$ model-to-observations slopes and correlation coefficients in the text as “As shown in Figure 3a, GEOS-Chem overestimates surface NO$_2$ mixing ratios at these sites by over 66%, compared with observations (slope=1.07; correlation coefficient=0.88).” (lines 170-171) and “Furthermore, the discrepancy for the differences of surface NO$_2$ mixing ratios over Europe between EURO_EDGAR and observations is further reduced (by 50%), relative to the control runs, with a model-to-observation slope of 0.92 and a correlation coefficient of 0.83 (Fig. 3b).” (lines 178-181)**

*5) Lines 206-208: Is the magnitude of the snowpack NOx reservoir depletion of right order to explain this? Is the source linearly dependent on the reservoir? Would it be hard to test this in the model to see if it improves the model bias? i.e. can you scale the monthly emissions according to this finding? Perhaps not necessary, but a brief discussion of the order of magnitude of depletion and how that relates to the model bias would be helpful.*

**Response: Thanks for the excellent questions. Snowpack nitrate photolysis plays an important role in affecting the surface NO$_x$ mixing ratios during late spring and summer over Summit, Greenland. Dibb et al. (2007) demonstrated that nitrate concentrations in the snowpack peaked in June and declined toward fall by ~ 1/3. Moreover, Van Dam et al.**

**(2015) offered the direct evidence that NO$_x$ mixing ratios within the snowpack showed declining trend from June to October, which may partially explain why we would see the declining trend of surface NO$_x$ mixing ratios over Summit from May-October. We have therefore added this discussions in the text "Dibb et al. (2007) reported that nitrate concentrations in the Summit snowpack peaked in June and declined toward fall by ~ 1/3. Van Dam et al. (2015) further showed decreasing trend for NO$_x$ mixing ratios within the snowpack at Summit from June to October. This may partially explain why we would see the declining trend of surface NO$_x$ mixing ratios over Summit from June toward fall. The NO$_x$ emissions from snowpack are affected by a number of factors including nitrate concentrations and solar radiation available and the responses can be very non-linear. Further investigations are needed to account for the seasonal variations of snowpack NO$_x$ emissions from nitrate photolysis in the model, i.e., constrained by seasonal snowpack NO$_x$ emission flux measurements in the future." (lines 209-217)**

*6) Does this model include the PAN budget updates from the Fischer et al. study that is mentioned? This should be stated clearly. Arnold et al., (2105) showed that GEOS-Chem produces less PAN relative to CO than other models in Arctic air masses influenced by fires. It would be useful to refer back to this here to give context to the model performance relative to that found for other models.*

**Response: Points are well taken. We have added clarification and discussion in the text - "For instance, a study by Fischer et al. (2014) showed improved agreement between modeled and measured PAN in the high latitudes when assignining a portion of the fire emissions in the model above the boundary layer and also directly partitioning 40% of NO$_x$ emissions from fires into PAN. We carried out a sensitivity test with similar treatments, but no significant improvements in the model simulated surface PAN were observed at the Summit site. Therefore, we did not include the PAN updates from Fischer et al. (2014) in other model simulations in this study." (lines 244-250)**
**We have also added discussion on the reference of Arnold et al. (2015) - "This is consistent with the study by Arnold et al. (2015), which reported that model simulated PAN mixing ratios in GEOS-Chem were lower than ARCTAS observations over high-latitude**

**atmosphere in the Arctic. Meanwhile, this study also revealed that GEOS-Chem produced less PAN relative to CO in Arctic air parcels that were influenced by fires, compared with other models." (lines 233-237)**

*Typographical / editorial corrections:*

*Line 58: ": : :while PAN mixing ratios were lower in fresh boreal fire plumes." This sentence in unclear. Lower than observed? Lower than in other air mass types simulated in the model?*

**Response: Thanks for pointing this out. Model simulated PAN mixing ratios were lower than the observations. Therefore, we have modified the whole sentence as "They found that model simulated NO$_x$ mixing ratios were higher than observations, while PAN mixing ratios were lower than the observations in fresh boreal fire plumes." (lines 60-62)**

*Line 82: ".. that the snowpack emits.."*

**Response: Done.**

*Paragraph beginning Line 141 contains mixed (past / present) tenses. Please adjust the text to make it consistent.*

**Response: Points are well taken. We have corrected the paragraph as "We first run the standard GEOS-Chem model with a-priori emissions and compare the simulation results against observations for various species (including NO$_x$, PAN, C$_2$H$_6$, C$_3$H$_8$, CO, and O$_3$, as shown in Fig. 1). Then we focus on the model-observation discrepancies, and where applicable, made revisions to the model simulations and further evaluate the improvement in model performance, as discussed in details below." (lines 139-143)**

*Line 156: " not observed in the data." Better to simply say ".. not observed".*

**Response: Agree. We have deleted "in the data" in the revised text.**

*Line 161: Omit word "mannually" (which should be spelled "manually" in any case).*

**Response: Typo has been corrected.**

---

## Author Comment (AC2) · 13 Oct 2017

Response to Dr. Hongyu Liu's comments:

We thank Dr. Hongyu Liu for his valuable and thoughtful comments. Our responses to the comments are provided below, with Dr. Hongyu Liu's comments italicized and our responses in plain and bold fonts.

*This paper presents a GEOS-Chem model analysis of surface ozone and its precursors (NOx, PAN, C2H6, C3H8, CO) observed at Summit, Greenland during the period of July 2008 - June 2010, with a focus on their concentrations and seasonal variations. Modeling tropospheric ozone in the Arctic has been challenging, and it is very interesting to use a state-of-the art chemical transport model to test and improve our understanding of its sources and variability. The authors identified the discrepancies between the GEOS-Chem simulations and observations, which were then examined using various model perturbation experiments. The results are original, and the paper is concise and very well written. I recommend its publication on ACP with some minor modifications, as itemized below.*

*1). Title - Using "tropospheric ozone" in the title is a bit misleading. Although this study also compared the model vertical profiles of ozone and specific humidity with ozonesonde observations, the main scope of this paper is "surface ozone".*

**Response: We agree with the reviewer's comment. We therefore change the title as "Surface ozone and its precursors at Summit, Greenland: comparison between observations and model simulations".**

*2). Section 2: It is not clear which version of the GEOS-5 meteorological data archive was used. Is it GEOS-5.1.0 or GEOS-5.2.0? See e.g., http://wiki.seas.harvard.edu/geos-chem/index.php/GEOS-5_met_field_reprocessing and "http://wiki.seas.harvard.edu/geos-chem/index.php/List_of_GEOS-5_met_fields".*

**Response: Thanks for pointing this out. It is GEOS-5.2.0. We have added this in Section 2 as "The GEOS-Chem model has fully coupled $O_3$-$NO_x$-VOC-Aerosol chemistry mechanism and is driven by assimilated meteorological data from the Goddard Earth Observing**

**System version 5.2.0 (GEOS-5.2.0) of the NASA Global Modeling Assimilation Office."**
**(lines 105-107)**

*3). Section 2, 2nd paragraph: "Time series data were archived with 3-hr temporal resolution at the Summit grid box" — I think you meant "grid column". Moreover, it is not clear how the model output was sampled in the vertical. The elevation of Summit is 3212m asl. Did you sample the model bottom layer, or the model vertical layer that is about 3212m above the sea level? The latter may very well be different than the former because the topography is not well resolved at coarse resolution. Would the results about model overestimates or underestimates found in this paper be different if the alternative way of model sampling is used (e.g., lines 206-207)?*

**Response: Good points. We archived the time series data with 3-hr temporal resolution at Summit grid box for each model vertical level, including the model bottom layer. For comparison with surface observations at Summit, Greenland, we sampled the data for the model bottom layer. Indeed, the topography is not very well resolved at coarse model resolution, and we believe diagnostics for the model bottom layer would work better than those for the 3212 m level for comparison to the surface measurements.**

*Minor editorial comments:*

*Line 43: change the "and" before "volatile organic compounds" to comma.*

**Response: Changed.**

*Line 56: the ARCTAS mission*

**Response: Corrected.**

*Line 66: What do you mean "O3 mixing ratios below the boundary layer"? Within the boundary layer?*

**Response: Yes, it is within the boundary layer. We have therefore corrected the sentence as "Wespes et al. (2012) also revealed that model simulated $O_3$ mixing ratios within the**

boundary layer were significantly underestimated during spring-summer, compared with ARCTAS measurements." (lines 68-70)

*Line 77: "....used to be the global default anthropogenic C2H6 emission inventory" - Do you mean "default" in GEOS-Chem or any other models?*

**Response: Yes, it is the GEOS-Chem default anthropogenic $C_2H_6$ emission inventory.**

*Line 228: add "over Greenland" after "PAN".*

**Response: Added.**

*Line2 268-270: "relative to NEI11_MIX" – isn't this relative to NEI11? Remove it?*

**Response: Well, it is relative to NEI11_MIX. In terms of emissions, you are right because we don't change the MIX emissions in this sensitivity simulation. Therefore, in order to avoid confusion, we give a simulation name for this sensitivity run and change this sentence as "We therefore run a sensitivity simulation by increasing the NEI11 $C_2H_6$ emissions by 40% and keeping other model configuration identical to NEI11_MIX (hereafter referred to as NEI11_40_MIX). We find this update leads to an increase in the model simulated annual mean surface $C_2H_6$ mixing ratios over Summit by only 6% during the period of 07/2008-06/2010 (figure not shown), still not able to explain the high model bias." (lines 283-287)**

*Line2 279-281: The annual mean agrees quite well with observations, but the simulation is worse in summer.*

**Response: Points are well taken. We have changed this sentence to "We find that the simulated annual mean surface $C_2H_6$ mixing ratios at Summit from NEI11_MIX20 agree quite well with observations (within 1%). Similarly, better agreement between model and observations are found for monthly average values for October - January. However, the new simulation is not able to reproduce the seasonal cycle of $C_2H_6$ - the model signficantly underestimates in February – April but overestimates in June – September (Fig. 5)." (lines 296-301)**

*Lines 283-288: Good point, but this long sentence needs a break.*

**Response: Thanks for pointing this out. We have divided the long sentence to "Note that this standard version of GEOS-Chem does not account for the sink of $C_2H_6$ from the reaction with chlorine, which could reduce the global annual mean surface $C_2H_6$ mixing ratio by 0-30%, and the global burden of $C_2H_6$ by about 20% (Sherwen et al., 2016). However, this may introduce additional uncertainty for our measurement-model comparison, together with the uncertainty in the seasonality of $C_2H_6$ chemistry." (lines 304-308)**

*Line 325: "Unfortunately, ..." – "However, ..."*

**Response: Corrected.**

*Lines 339-340: "..., which implies that GEOS-Chem possibly underestimates STE for O3 over Summit" – This is interesting and appears consistent with Choi et al., ACP 2017 (https://www.atmos-chem-phys.net/17/8429/2017/ , see their Fig. 6) , where the GMI CTM driven by MERRA (GEOS-5.2.0) underestimates ozonesonde-observed ozone in the Northern Hemisphere high-latitude upper troposphere.*

**Response: Thank you for providing us a reference source, which also attributed the model low bias to STE. We have included a discussion about this study in the text as "This is consistent with the study by Choi et al. (2017), which found low bias with model simulated $O_3$ mixing ratios over high-latitude upper troposphere of the Northern Hemisphere, compared with ozonesonde data, and attributed the low bias to weak STE in the model." (lines 359-361)**

*Lines 358-359: Summit, Greenland; surface ozone*

**Response: Good suggestion. We have changed the whole sentence to "We combine model simulations with two-year (July 2008 - June 2010) ground based measurements at Summit, Greenland, to investigate the abundance and seasonal variations of surface $O_3$ and related species in the Arctic." (lines 370-372)**

*Figures 2, 3,5, S1: In the caption, state briefly what the perturbation simulations are*

*and refer the reader to the text for details.*

**Response: We have added additional descriptions in the captions for Figures 2, 3, 5, and S1. Please refer to our revised manuscripts for details.**